# Cooperative but Dependent–Functional Breed Selection in Dogs Influences Human-Directed Gazing in a Difficult Object-Manipulation Task

**DOI:** 10.3390/ani14162348

**Published:** 2024-08-14

**Authors:** Péter Pongrácz, Csenge Anna Lugosi

**Affiliations:** Department of Ethology, ELTE Eötvös Loránd University, 1117 Budapest, Hungary; lugosipolly@gmail.com

**Keywords:** working dogs, functional breed selection, unsolvable task, ‘looking-back’ response, persistence

## Abstract

**Simple Summary:**

The main components of dogs’ species-specific behavior are thought to be mainly influenced by domestication. This implies that these phenotypes are expected to be present almost uniformly across the vast number and variety of dogs. However, the evolution of dogs did not end with domestication, as humans subsequently selected them for distinctly different tasks and levels of interactivity with their handlers. We hypothesized that ‘cooperative’ dog breeds would also show higher levels of dependency toward humans in a difficult problem-solving task than those dogs that were selected for being independent problem solvers. Our target behavior was ‘looking back at the human’, which is a typical reaction of socialized dogs to a difficult task. Indeed, cooperative dogs performed more gaze alternations between the reward and the nearby humans, and they looked back more than the independent dog breeds did. Importantly, the results cannot be explained by different levels of persistence between the breed types. Functional selection in the recent past of dogs can be considered as an excellent basis for biologically relevant explanations for the breed-level variability in dog behavior.

**Abstract:**

It is still largely unknown to what extent domestication, ancestry, or recent functional selection are responsible for the behavioral differences in whether dogs look back to a human when presented with a difficult task. Here, we tested whether this ubiquitous human-related response of companion dogs would appear differently in subjects that were selected for either cooperative or independent work tasks. We tested *N* = 71 dogs from 18 cooperative and 18 independent breeds. Subjects learned in a five-trial warming-up phase that they could easily obtain the reward from a container. In trial six, the reward became impossible to take out from the locked container. When the task was easy, both breed groups behaved similarly, and their readiness to approach the container did not differ between the last ‘solvable’ and the subsequent ‘unsolvable’ trial. Task focus, looking at the container, touching the container for the first time, or interacting with the container with a paw or nose did not differ between the breed groups, indicating that their persistence in problem solving was similar. However, in the ‘unsolvable’ trial, cooperative dogs alternated their gaze more often between the container and the humans than the independent dogs did. The frequency of looking back was also higher in cooperative dogs than in the independent breeds. These are the first empirical results that suggest, in a balanced, representative sample of breeds, that the selection for different levels of cooperativity in working dogs could also affect their human-dependent behavior in a generic problem-solving situation.

## 1. Introduction

So far, the most promising approach to the investigation of species-specific behavioral characteristics of dogs involved experiments where the dogs’ behavior was compared to the responses of age-matched socialized (tame) wolves (e.g., [1]). In these studies, it became clear that domestication could cause changes in the timing and capacity of dogs in developing specific preferences, bonds, and communicative skills with humans. In the strange situation test that was modified by Topál and colleagues [2] for testing the dog–owner bond, dog puppies were more responsive toward their owner than toward an unfamiliar human participant, whereas extensively socialized wolves did not show differential interest toward their handler and an unknown person. According to the authors, the attachment system of the dog probably became specifically human-oriented due to their domestication [3], although there are indications that wolves might also show signs of intra- and inter-specific attachment bonds [4,5]. Dog puppies were also found to be capable of following human visual communicative signals at a younger age than tame wolves [6]. Additionally, compared to wolves, dogs seemingly lost most of their cooperative inclinations with their conspecifics; however, they became highly dependent on humans in their cooperative behaviors [7]. In a string-pulling task, both dogs and wolves cooperated successfully with the human partner; however, wolves also keenly cooperated with their conspecific partner, while dogs did not really cooperate with other dogs [8]. Beyond the mutually cooperative tasks (e.g., [9]), dogs usually show a strong dependency on humans in a wide array of problem-solving scenarios when they have the opportunity to interact with humans. Among others, dependency on humans can manifest itself in reliance on human encouragement [10], ostensive communicative signals [11], or observing the behavior of humans as demonstrators (e.g., [12]).

One of the classic and fundamental findings highlighting the dependency of dogs on human assistance is the robust ‘looking-back’ response (LBR) toward a nearby human (typically the dog’s owner) when the dog encounters a difficult (e.g., detour task [13]) or even impossible task (e.g., obtaining food from a blocked dispenser, see [14,15] for review). Originally, looking-back behavior directed at humans was considered a species-specific (i.e., domestication-related) feature, because juvenile dogs started to look at the human readily after they faced difficulty, while similar-age socialized wolves (and companion cats [16]) kept trying to solve the task on their own, without looking toward the human [17]. Later, it was found that in the case of a difficult (often dubbed as ‘unsolvable’) task without the presence of humans, wolves were still more motivated and persistent to obtain the reward [14]. The mechanism behind the LBR has multiple explanatory approaches, including the one that assumes that subjects with higher persistence will refrain from the LBR longer than individuals with lower task-related persistence [18,19]. Other authors consider looking-back behavior as a request for help and a sign of dependency on humans [20].

Nevertheless, in general, companion dogs are almost uniformly prone to perform LBR in difficult task situations [21]. Among the potential influencing factors, previous studies have shown that individual experience, age, keeping conditions, and training level of the dogs can affect the manifestation of LBR. Brubaker et al. [22] found that free-ranging dogs looked less toward the nearby human, while shelter and pet dogs showed equally more gazing behavior. Persson et al. [23] showed that older dogs gazed sooner and more often at the human than younger dogs did. This suggests that it may take time until young dogs gain experience that looking at humans often ends with a positive outcome [21]. D’Aniello and Scandurra [24] reported that dogs living in the kennels of a ‘dog museum’ that provided limited human contact showed less LBR and with a higher latency than pet dogs did, regardless of the training level of the latter. As for the effect of the training level of dogs, according to Marshall-Pescini et al. [25], untrained pet dogs spent significantly more time looking back at people than trained dogs did (e.g., with formal training and competition experience in either agility, Schutzhund, or search and rescue). This result emphasizes that whether the main reason behind LBR is ‘dependency’ or a low level of persistence, regular work-related training can weaken the effect of both factors, consequently reducing the amount of LBR. 

Apart from investigating the effect of the various environmental and ontogenetic confounders, breed-related differences are also at the forefront of dog behavior research regarding LBR. Taking into consideration the various pressures of selection in the history of dogs, one could assume that various dog breeds would also show more or less LBR when they face difficulties in problem solving. Ujfalussy and colleagues [26], for example, found that brachycephalic breeds showed more LBR in a difficult task than normocephalic breeds did. According to their explanation, this result may indicate that brachycephalic dogs were selected for a more human-dependent predisposition, which could also explain their vast popularity among companion dog owners. However, this study, similar to others (e.g., [27,28]), had its limitations because of the low number of investigated dog breeds. Ujfalussy and colleagues tested only two brachycephalic dog breeds (English and French Bulldogs) and compared their behavior to a single herding dog breed, the Mudi. It would be hard to imagine that the only factor that differentially affected the LBR in these dog breeds was their head shape. Using only a handful of breeds per cluster can result in problems that unavoidably limit the scope of conclusions. No matter how well the given dog breed represents a type or group, there will be additional factors that confound the results, such as the dog breeds’ original function, genetic relatedness, typical purpose, and keeping conditions [29]. Another potentially problematic issue is when a larger amount of dog breeds is included in a study without a solid a priori hypothesis. Such studies usually concentrate on a convenience sample of the most available popular breeds instead of breed groups (e.g., [30,31]). For example, Junttila and colleagues tested an impressive number of subjects belonging to 13 popular dog breeds. They found breed differences in tasks measuring social cognition, problem solving, and inhibitory control, and they concluded that these traits could be subject to different artificial selection pressures in the various dog breeds. This is an interesting result that should be taken into consideration when we investigate breed groups. However, this method is only suitable for explorative research, and no broader ecologically relevant conclusions can be drawn from these breed-specific explanations because of the lack of an overarching theoretical research hypothesis.

More promising are the investigations where the breeds’ genetic relatedness is used as an explanatory variable (e.g., [32]). However, as these papers naturally provide results that can be explained by the genetic distance between dog breeds (e.g., compared to the ‘wolf-like’ ancestor), in turn, they shed less light on what sort of selection could result in the found similarities and differences. In other studies, ancestry and function are mixed somewhat in the formation of experimental groups (e.g., [33,34]), which is useful for defining differences, but makes it difficult to separate the potential factors behind them. Passalacqua et al. [35] worked with three groups of breeds: Primitive, Hunting/Herding, and Molossoid, in which dogs in the Hunting/Herding category significantly showed the most gazing behavior. The authors concluded that selection for cooperative work with humans may have had a greater influence on this behavior than the subjects’ genetic relatedness to wolves.

An even more promising approach would be when such grouping variables are chosen that relate to a more-or-less independent selective process from the genetic clades of dog breeds. Functional breed selection in the more recent (post-domestication) past of many dog breeds offers a successful lead for the investigation of behavioral phenotypes that relate to dog–human interactions. This approach does not focus on the specific working task of the dogs but concentrates on how they are supposed to do it. There are two main types of working dogs from the aspect of their function during joint effort with their human handler: dogs that work in cooperation with, or independently from, their human partner [12,36]. For cooperative dogs, it was adaptive to pay attention to the regular visual and acoustic cuing from their handler and execute their instructions without hesitation. In the case of independent working dogs, they were selected to make decisions and solve problems on their own during their work. It was found that the two types of working dogs performed equally when the task did not involve human contribution; however, cooperative breeds were more successful when they had to rely on human communicative signals [36]. In contrast, independent dogs showed stronger reward-maximizing tendencies [9] and visited the ambiguous reward location more readily in a cognitive bias scenario [37]. Furthermore, the independent breeds more frequently tried to steal forbidden food when their owner was not aware of it [38]. Lugosi and colleagues [39] found that independent dogs learned better from a conspecific demonstrator in a ‘classic’ detour paradigm, while cooperative dogs only benefited from observing a human demonstrator [12]. Remarkably, cooperative dog breeds also showed more LBR in the study of Lugosi et al. [39], coinciding with their larger difficulties in mastering the detour task. On the other hand, Lazarowski et al. [40] and Hirschi et al. [20] compared cooperative and independent dog breeds in terms of LBR during an impossible task, and they did not find differences between the groups. However, Hirschi and colleagues only included terriers and herding dogs in their study, and thus the representativeness of the two working dog types can be considered as somewhat skewed. Lazarowski and colleagues [40] tested a large number of Labrador Retrievers along with a few specimens of other cooperative breeds, while only a few independent breeds were included (meanwhile, the authors invited mixed-breed dogs as well as hard-to-classify breeds to their sample). We suggest, therefore, that for investigating the effect of functional working breed selection on the dependency of dogs on human assistance, a much clearer and more representative choice of breeds would be needed.

### Aims of Our Current Study

In this paper, we hypothesized that functional breed selection could have an effect on the dependency of dog breeds on human assistance in a difficult (impossible) task context. We predicted that when the task became difficult, independent dogs would show higher persistence/lower dependency that would result in lower-frequency and longer-latency LBR, compared to what we would see in cooperative dog breeds. An alternative hypothesis could be that dependency on human assistance is a more basic/ancient feature in dogs, evolved soon after domestication, and thus the subsequent functional breed selection would not affect dogs’ LBR. In this case, we would expect no significant difference in LBR durations and frequencies between the two working dog breed types.

## 2. Materials and Methods

### 2.1. Subjects

We tested *N* = 71 adult (1 year or older, mean ± SD = 4.12 ± 2.42 years) companion dogs in the presence of their owner. To categorize the dog breeds according to their original work functions, we followed the method that was used in a number of earlier publications (e.g., [6,9,12]). Only purebred dogs were recruited that belonged to either the cooperative working breeds or to the independent working breeds. We considered any dog breed that is recognized by one or the other major kennel clubs (FCI, AKC, and KC). Only those breeds were used where their function type (independent or cooperative) was possible to be unambiguously determined according to the historical breed description, accessible from the official breed standards. We intentionally avoided the inclusion of those breeds for which the original work function has long been removed from the goals of breeding (such as the English Bulldog), or if the breed clearly had a ‘toy/companion’ designation (such as the Pekingese or the Bolognese). Importantly, the test groups were assembled from multiple breed representatives, where no more than 4 dogs from the same breed were tested per group. We tested 18 breeds from the independent and 18 breeds from the cooperative working dog groups. Each subject was tested only once. We provide the basic demographic details of the subjects in Table 1. Furthermore, we collected data about their keeping conditions (indoor only, indoor–outdoor, and outdoor only), as well as the level of training the dogs received (none, training at home, course at dog school, regular dog school, private trainer, and specific sports/work training).

### 2.2. Equipment

We ran all tests between September and December 2023. All tests were performed outdoors, on a grassy, open area connected to the Eötvös Loránd University campus. The equipment was a 13.5 × 13.5 × 7 cm (0.7 L) sealable, transparent plastic container. The lid of the container was fixed onto a 33 × 33 cm wooden board. The container was either openable for the dog (when the body of the container was not locked onto the lid), or it was set as unopenable (when the body of the container was secured by the lid’s locking clamps). The latter condition was the ‘difficult task’. The starting point was 2 m away from the container. We placed the reward in the container, which was a favorite toy or food for the dog, selected and brought by the owner. We recorded the tests with a video camera (BLOW Go Pro4U) that was positioned on a tripod and placed in the midline between the box and the starting point, 3 meters away from the testing area. 

### 2.3. Test Groups

Each subject participated only once in the investigation. We assigned the dogs to the groups by paying careful attention to the balanced distribution of sex, age, keeping condition, and training level of the subjects. The following test groups were formed, with the number of subjects in each group that were used for statistical analyses presented in parentheses (i.e., Ns after exclusions):

Independent dogs (*N* = 35)

Cooperative dogs (*N* = 36)

We determined the desired sample size by using the equation for finite populations: ń=n1+z2×p^1−p^ε2 N
where *z* (*z*-score) = 1.96 for the 95% confidence level, *ε* (margin of error) = 0.05, and p^ (population proportion) = 0.50. We expected that the population of suitable dogs (N) for our test would be 80 (based on previous social media subject recruiting campaigns and a reasonable timeframe, we could invite no more than 80 subjects that belonged to the targeted breed groups). The calculated sample size was *N* = 67. We actually had the opportunity to test a slightly higher number of subjects (*N* = 71 + 6), expecting that some of the subjects would need to be excluded for various reasons (lack of motivation, technical issues, and non-complying owner). The exact details of exclusions and the actual number of excluded subjects are provided in the next section.

### 2.4. Testing Procedure

The dogs were provided with a task in which they had to obtain a reward from the container. In the familiarization phase (5 trials), the container was easy to open, because the lid was not fastened to it. In the test phase (1 trial), the lid was fastened to the container, and thus the dog was able see the reward through the transparent wall of the container, but it could not open the container. Our goal was that dogs would readily approach the container and take the reward out toward the end of the familiarization phase, and then the task would become impossible to solve for the single-trial test. 

After arriving to the testing site, dog owners entered the testing area with the experimenter (who was always the same woman, C.A.L.). The experimenter explained the procedure to the owner, including what to do and what not to do during the test. The dog was allowed to explore the area on a leash, but it was not allowed to go to the container. We asked the owner to keep the dog on the leash and position themselves at the starting point. The owner had to stand directly behind the dog, and both faced the container. The experimenter called the dog’s attention (by saying its name and, for example, ‘look’) and then walked to the container, conspicuously holding a piece of food (or the toy) in her hand, which she then put in the container. The experimenter showed her empty hands to the dog, then she returned to the starting point and stood about 1 m behind the owner. After this, the experimenter told the owner to release the dog. The owner was also told to encourage the dog to solve the task (i.e., “bring it”, “get it”). We tested every subject in six consecutive trials, with no extra delay before the last (test) trial. In the familiarization phase (5 trials), the container was easy to open, because the lid was not fastened to it. The dog had 60 s to solve the task—if it obtained the reward from the container within this time limit, the owner had to recall the dog to the starting point, and the next trial started. If the dog did not get the reward in 60 s, the trial ended, and the owner had to call the dog back to the starting point. In the test phase (1 trial), the lid was fastened to the container, and thus the dog could see the food/toy through the transparent wall of the container, but it could not open the container. The dogs had 3 minutes to try and obtain the reward. In most cases, dogs gave up earlier and the trial ended sooner than 3 min. A dog was considered as giving up if it did not go closer than 1 m to the container for 30 s. 

### 2.5. Exclusions

We excluded those subjects that were not motivated to perform any trials or lost interest for further performance during the test. A dog was considered to have lost interest if it did not approach the container upon its release from the starting point or only approached once. Altogether, we had to exclude *N* = 6 dogs (3 cooperative and 3 independent dogs) for this reason.

### 2.6. Behavioral Variables

Each test was video recorded. We used Solomon Coder (beta 19.08.02, Copyright by András Péter) for the extraction of data from the video sequences. Table 2 shows the behavioral variables we used for the analysis.

### 2.7. Statistical Analyses 

All statistical analyses were performed with the SPSS (version 29) software (IBM, Armonk, NY, USA). Raw data, used for statistical analysis can be accessed in the Electronic Appendix A.

Generalized Linear Models were used for the analysis of task focus relative duration, as well as for the relative duration of looking at the container and interacting with the nose or paw. Furthermore, GzLM (Generalized Linear Model) was also used for the analyses of frequencies of owner encouragement, LBR, and two-stage gaze alternation. Breed group, dog’s sex, reproductive status, keeping conditions, and training level were used as factors. We also included the biologically relevant two-way interactions (breed group x training level and keeping condition) in the model. We applied the Benjamini–Hochberg correction against the inflation of Type I errors due to multiple comparisons, and we reported the adjusted *p*-values in the case of these analyses. 

First-touch latency was analyzed with the Cox regression model. We ran between-group comparison, because in this way, we could assess whether the dogs in the two breed groups were equally motivated to solve the task. Besides the breed group, keeping conditions and training level were also added to the model. 

Finally, we analyzed the first-touch latencies with General Linear Mixed Model, where trial was the repeated factor, and we added breed group as an independent variable to the model. We also added the two-way interaction of trial and breed group to the model.

Whenever we analyzed the two-way interactions, we performed a backward model selection, removing the non-significant interactions. In each instance, results of the final (simplest) model were reported. The α level was 0.05. 

To ensure that the coding of the behavioral variables was reliable, a second coder coded the video footage of 15 subjects, and we performed Spearman correlation analysis for the frequency variables. The coding proved to be reliable in the case of each variable: Trial 5 first-touch latency (R_15_ = 0.958; *p* < 0.001); Trial 6 first-touch latency side-switching (R_15_ = 0.969; *p* < 0.001); looking-back frequency (R_15_ = 1.000; *p* < 0.001); two-stage gaze alternation frequency (R_15_ = 1.000; *p* < 0.001); task focus duration (R_15_ = 1.000; *p* < 0.001); owner encouragement frequency (R_15_ = 1.000; *p* < 0.001); Trial 5 looking at the container duration (R_15_ = 0.742; *p* = 0.002); Trial 6 looking at the container duration (R_15_ = 0.982; *p* < 0.001); Trial 5 nose touch frequency (R_15_ = 0.677; *p* = 0.006); Trial 6 nose touch frequency (R_15_ = 0.808; *p* < 0.001); Trial 6 paw touch frequency (R_15_ = 0.990; *p* < 0.001).

## 3. Results

We compared the 5th trial of the familiarization phase and the test phase (6th trial) with regard to the latency of the dogs’ first touch on the container, which did not show significant association with the trial (Mixed GLM, F(1, 69) = 0.079; *p* = 0.780), the breed group (F(1, 69) = 2.752; *p* = 0.102), or the interaction between them (F(1, 69) = 0.272; *p* = 0.603). 

Most of the analyzed behavioral variables did not occur (or occurred only sporadically) in the 5th trial of the familiarization phase. For example, dogs did not look back to the owner/experimenter, did not show gaze alternations, and they barely used their paw when interacting with the container. They did not leave the vicinity of the container, and probably because of the ease of the task, the owners did not encourage them. Therefore, we only analyzed those parameters in the 5th trial that occurred in a relevant frequency, to see whether the two breed groups showed different interactions with the container when the task was easy to solve. In the familiarization phase, we did not find significant associations between any of the factors and the duration of looking at the container or the dogs’ interaction with the container with their nose (Table 3).

In the test phase (Trial 6), we found a significant association between breed groups and frequency of LBR (Table 4). Cooperative dogs looked back to the owner more often than the independent dogs did (Figure 1). We did not find significant associations between the frequency of LBR and the other factors: sex, reproductive status, keeping conditions, and training level (Table 4). 

Two-stage gaze alternation (TSGA) showed a significant association with the breed group (Table 5). Cooperative dogs showed significantly less TSGA than the independent breeds did (Figure 2). The other factors did not affect TSGA (Table 5).

First-touch latency in the test phase was significantly associated with the keeping condition: indoor-only dogs touched the container sooner than dogs with outdoor access. The breed group and training level of the dogs did not have a significant effect (Table 6).

In the case of interacting with the container with paws, we found a significant interaction between the breed group and the dogs’ keeping condition (Table 7). Indoor-kept independent dogs used their paws more than those independent dogs that had access to the outdoors. There was no such keeping-related difference between the cooperative dogs (Figure 3). Other motivation-related behaviors (nose–container interaction, task focus duration, and owners’ encouraging utterances) did not show significant associations with any of the fixed factors (Table 7).

## 4. Discussion

We applied a well-known ‘difficult task’ paradigm on two functionally different groups of working dog breeds, to investigate whether artificial selection for a more cooperative or independent working style with humans could result in differences of persistence and human orientation. Our results showed that when the task was easy to solve, both breed groups behaved similarly, and their readiness to approach the food container did not differ between the last ‘solvable’ and the subsequent ‘unsolvable’ trial. However, the breed groups showed a remarkable difference in their occurrence of looking back to the nearby standing human when the task was difficult to master. Cooperative dogs alternated their gaze more often between the container and the humans than the independent dogs did. The frequency of looking back at the humans was also higher in the cooperative dogs, while the independent dogs showed a lower frequency of looking back. The behavioral variables that indicated the dogs’ motivation and persistence in problem solving (task focus, looking at the container, touching the container for the first time, and interacting with the container with paw or nose) showed no association with the breed group assignment of the dogs. The only exception was the relative duration of paw usage, where indoor-only independent dogs were more likely to use their paws than the independent dogs that also had outdoor access. We found no effect of confounders, such as dogs’ sex, neuter/spay status, or their training level. With regard to keeping conditions, indoor-only dogs touched the container sooner in the test phase than the dogs with outdoor access. This last result was independent of the breed group designation of the subjects. The original tasks of many breeds included the need for touching/grabbing of live or dead animals, and these breeds were represented in both breed groups (e.g., retrievers, pointers, and utility breeds in the ‘cooperative’ group, and terriers and sighthounds in the ‘independent’ group). We could, therefore, speculate that the difference between the first-touch latencies of subjects with regard to their keeping conditions could be connected to the various levels of novelty they normally experience at home. As dogs with free access to the outdoors probably encounter more novel stimuli on a regular basis, they may react with less enthusiasm to the objects provided during a test. Indoor-only dogs, on the other hand, could take the test with the container and reward as a case of environmental enrichment and react with higher motivation. Another explanation could be that the owners of indoor-only dogs may provide their dogs with puzzle games more often than the indoor–outdoor dogs’ owners do. Therefore, indoor-only dogs could be more familiar with these types of problem-solving tasks, and they approached the container faster in our test, too.

According to our main hypothesis, we expected that as a result of different task-related selection pressure in the past, cooperative dogs would show more human-directed gazing than independent dogs when the task became difficult to solve. The results supported this prediction, as the cooperative dogs alternated their gaze between the container and the nearby humans more often than the independent dogs did. Gaze alternation can be considered as an especially good indicator of communication intent [41], as it has a visual-attention-eliciting component compared with a ‘simple’ looking-back response. It was found earlier that similar to human infants, dogs use more gaze alternation when a task becomes difficult, and their behavior was even in association with the attentional stance of their human audience [42]. In the present experiment, the human participants (owner and experimenter) could both be considered as ‘attentive’ during the trials; thus, our results showed an intriguing difference between the gaze-alternating behavior in the two functional breed groups. 

We also expected that the independent breeds may show higher persistence in the ‘impossible’ task phase, and this could coincide with their lower willingness to look back at the humans. Although independent dogs truly showed less frequent gaze alternation and looking-back responses, we found no indication that they would be more persistent/motivated to interact with the container than the cooperative dogs were. Earlier, it was found that independent dog breeds were keener in reward-maximizing (i.e., they showed a more ‘optimistic’ stance in a cognitive bias test [37]), which would predict that they also would be more persistent in the ‘unsolvable’ phase of this test. However, we found no evidence of either a more effective problem-solving performance or higher persistence of the independent breeds in the detour paradigm [12]. Persistence was found to be higher in the ‘ancient’ (or more ‘wolf-like’) breeds than in the more modern breeds [28], and wolves were found to be more persistent than dogs [43]. At the same time, ancestry (i.e., genetic distance from the wolf-like ancestor) does not associate with functional breed selection. This means that dog breeds that were selected for cooperative or independent working tasks can be found even in the same genetic clades [12,44]. This could be an explanation for why we did not find higher persistence in the independent dog breeds compared to the cooperative breeds during the ‘unsolvable’ phase of the test.

In the case of LBR, we found similar results to that described in gaze alternations: cooperative breeds looked back at the humans more frequently than the independent dogs did when the task became difficult (Trial 6). This association was not influenced by any other factor in our study, as probably one could expect that the training level of the dogs could have an effect here. For example, it could be assumed that higher levels of training would actually ‘mask’ the effect of breed-group-typical behavioral characteristics. For example, more elaborate training efforts would lessen the dependency of the cooperative dogs; however, when applied to independent dogs, training can enhance the dependency bond with the trainer. By testing Labrador Retrievers and Labrador x German Wirehaired Pointer crosses in a detection dog training program, Lazarowski and colleagues [40] found that higher levels of LBR in a difficult task at 11 months of age were predictive of future suitability for becoming a service dog, which highlighted the role of dependency in joint activities with humans where intensive training is involved. In the same study, dogs’ persistence in the difficult task was not associated with their subsequent success as a service dog. 

With regard to our results, we found no association between the training level of dogs and behavioral variables that are connected to the persistence of dogs during problem solving (e.g., task focus, looking at the container, and physical interactions with container). Cavalli and colleagues [45] found that more trained dogs showed higher persistence in a gazing and problem-solving task. However, as we did not find any stronger indications of different levels of persistence in the two breed groups’ behavior, we think that the first explanation (‘dependency’) would be a better fit to the different frequencies of LBR in the two breed groups. Based on our results, the association with functional breed selection could help to elucidate the evolutionary background of LBR in difficult task scenarios. In parallel with this, it warrants the need of careful (representative) sampling of dog breeds when this paradigm is used, as it is likely that the cooperation/independence of particular breeds can act as a powerful confounder here [29].

Previous studies have shown that domestication caused important changes in dogs’ capacity to develop specific preferences, bonds, and communicative skills with humans [1,8]. It would seem conceivable that dogs in general became dependent on human assistance soon after, and because of, their domestication. As a consequence, one could assume that human-dependent behaviors are generally and evenly present in any socialized dog. However, studies with free-ranging and shelter dogs have shown that lifetime experience with humans can have an effect on various human-directed behaviors (e.g., gazing [22] and point-following [46]). Ancestry also seems to be an influencing factor, where more ‘ancient’ dog breeds may show less intense human-directed behaviors (e.g., gazing [32]). There is a growing body of evidence that functional breed selection could also have a strong influence on human-related interactions of dogs. Cooperative breeds were selected to perform their task under the continuous control and guidance of their handler, and they were indeed more successful in tasks when they had to rely on human communicative signals [36]. Unlike independent dogs, they benefited from observing a human demonstrator in the ‘classic’ detour paradigm [12]. However, independent dogs outperformed cooperative breeds when instead of a human, they had to rely on their conspecifics: they learned better from a conspecific demonstrator [39]. In that study, cooperative breeds looked more at the owner, in parallel with their more pronounced difficulties in mastering the detour task. Our results are in line with these findings and confirmed the assumption that in some cases, functional breed selection may have modified the consequences of domestication on dogs’ capacity/willingness to interact with people. We can hypothesize that the artificial selection for more cooperative or more independent-acting dogs pushed the two main types of working breeds toward the two opposite ends of the dependency scale. As a result, cooperative dogs became more dependent on humans, while dogs selected for independent work prefer to solve problems on their own. 

Investigations targeting dog-breed-related behavioral differences are often plagued by confounders (experience with humans, keeping conditions, and training levels of the subjects) and biased/unbalanced sampling of dog breeds [29]. In this study, we placed great emphasis on testing as many dog breeds as possible from both groups, and we carefully avoided the over-representation of any breed in the sample. Within the two groups, we tried to cover the widest range of special work tasks, such as different types of herding, hunting, and sled dogs. Since we tested only purebred family dogs, we minimized the effect of uneven lifetime experiences with humans. We made sure that dogs from different keeping conditions, with different ages and training levels, were similarly distributed between the groups. Although we would expect less LBR from the more wolf-like dogs [28,34], ancient and modern breeds were similarly included in the cooperative and independent groups, so the genetic distance from the wolf-like ancestor was less likely to be a crucial factor here. Besides the ecologically relevant approach, these cautious preparations of the breed-sampling could also strengthen our interpretation of the results [29].

We acknowledge some limitations of the study; for example, that more specific task-related differences could also occur within breed groups, which may influence the results. For example, when paw use was found to be associated with keeping conditions within the independent breeds, a closer look at our sample showed that these indoor-only independent subjects belonged to underground hunter breeds, which can show more pronounced digging (pawing) behavior. These inequalities caused by the specific work-related behaviors should ideally be eliminated by a more inclusive and balanced breed composition at the sampling level; however, this would require a very high number of subjects. Another limitation is that we sorted the various working dog breeds into the ‘independent’ or ‘cooperative’ groups based on their original task, taken from the official breed standards. However, recent preferences in dog breeding for either more family-friendly or work-oriented behavior traits has resulted in the development of genetically and behaviorally different [47,48] ‘lines’ within some breeds (e.g., Retrievers, Border Collies, Australian Kelpies, etc.). As we used only a few specimens from each breed in our study, we could not balance between the potentially different within-breed variants, although based on our recruiting efforts, we can assume that our subjects were family companion dogs. Finally, while our sample was representative and balanced from the aspect of cooperative and independent working dog breeds, with the limited number of subjects, it was not possible to provide a large enough subsample of each combination of confounders (sex, reproductive status, keeping, and training conditions). As our main goal was to test the potential association between functional breed selection and the looking-back response of dogs in a difficult task, we aimed at having a similar distribution of confounders across the two breed groups. 

## 5. Conclusions

The division of dogs into two main categories based on functional breed selection provides an ecologically relevant approach and an overarching system for behavioral comparisons, regardless of the genetic relatedness or popularity of dog breeds. Based on our results, this work-related artificial selection for tighter or looser cooperation with humans could modify the general effects of domestication with regard to dogs’ dependency on humans. For the cooperative breeds, normally, it is adaptive to expect instructions from humans, and probably the task-related pressure of selection in the past made them, in general, more dependent on humans. An indirect indication for this could be the more intense occurrence of separation-related problems in cooperative dog breeds [49]. In contrast, for independent dogs, it was indispensable to concentrate on the task amidst their original working environment and make decisions without the owner. Interestingly, we found no indication that the independent breeds would be more persistent/motivated to interact with the container than the cooperative dogs were. Our results suggested that the looking-back response/gaze alternation can be considered as a manifestation of dependency rather than a lack of persistence. In the future, it would be interesting to investigate whether the stronger dependency of cooperative dog breeds on humans actually would enhance or lessen their work efficiency in realistic circumstances, such as scent detection work. Stronger dependency could mean higher sensitivity to human-given cues, which can theoretically lead to unwanted bias in scent-detecting scenarios [50].

## Figures and Tables

**Figure 1 animals-14-02348-f001:**
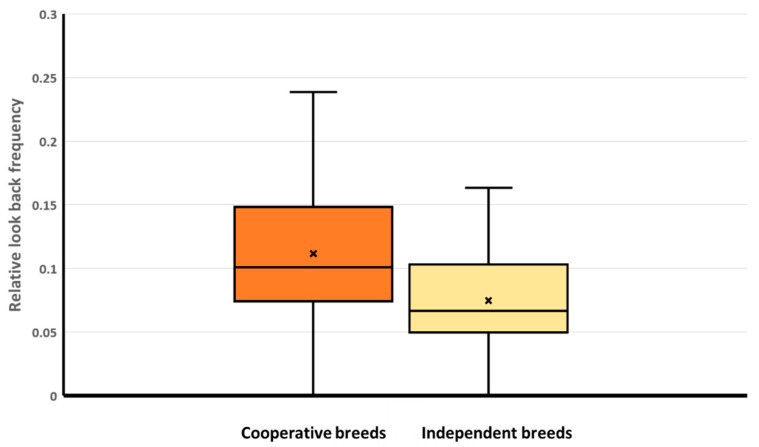
In the test phase (Trial 6), relative looking-back frequency had a significant association with the breed groups. Cooperative dogs looked back to the owner more often than the independent breeds did. Whiskers: minimum, maximum; box: Q1–Q3 interquartile range; horizontal line: median; x: mean. Darker brown color: cooperative breeds; lighter sand color: independent breeds.

**Figure 2 animals-14-02348-f002:**
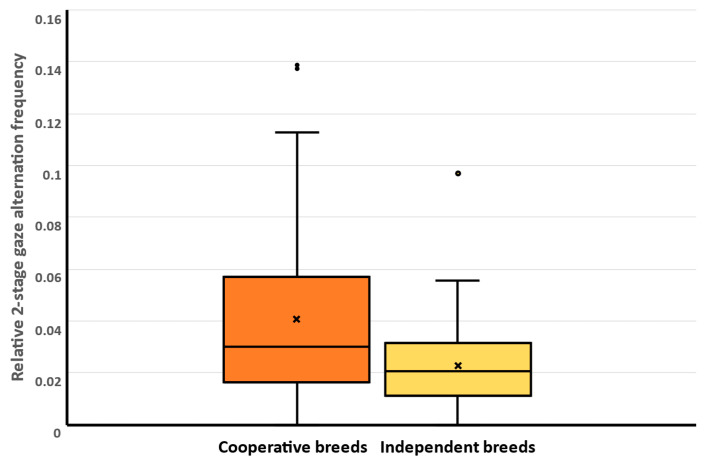
In the testing phase (Trial 6), the relative frequency of two-stage gaze alternation (TSGA) showed a significant association with the breed group assignment of the subjects. Whiskers: minimum, maximum; box: Q1–Q3 interquartile range; horizontal line: median; x: mean. Round circles above boxplots are the outliers. Darker brown color: cooperative breeds; lighter sand color: independent breeds.

**Figure 3 animals-14-02348-f003:**
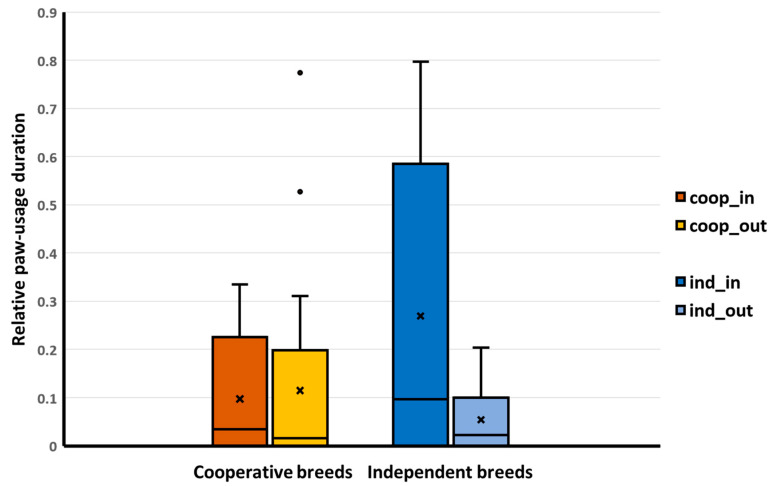
The interaction between breed group and keeping condition had a significant association with the relative duration of dogs’ paw usage on the box in Trial 6. Legend: coop_in, cooperative breeds, indoor-only; coop_out, cooperative breeds, indoor with outdoor access; ind_in, independent breeds, indoor-only; ind_out, independent breeds, indoor with outdoor access. Whiskers: minimum, maximum; box: Q1–Q3 interquartile range; horizontal line: median; x: mean. Round circles above boxplots are the outliers.

**Table 1 animals-14-02348-t001:** List of the subjects. We indicated their breed type (independent or cooperative group), their sex and reproductive status, and their age (in years).

Subject ID	Breed	Group	Sex	Reproductive Status	Age
1	English Cocker Spaniel	cooperative	female	spayed	7
2	English Cocker Spaniel	cooperative	male	intact	3
3	English Cocker Spaniel	cooperative	female	spayed	2
4	Australian Shepherd	cooperative	male	neutered	4
5	Australian Shepherd	cooperative	male	neutered	8
6	Australian Shepherd	cooperative	female	intact	1.5
7	Border Collie	cooperative	male	neutered	4
8	Border Collie	cooperative	male	intact	1.5
9	Border Collie	cooperative	female	spayed	3
10	Briard	cooperative	male	intact	2
11	Briard	cooperative	male	intact	5
12	Briard	cooperative	female	intact	2
13	Golden Retriever	cooperative	female	intact	4
14	Golden Retriever	cooperative	female	intact	1.5
15	Golden Retriever	cooperative	male	neutered	3
16	Dutch Shepherd Dog	cooperative	female	spayed	7
17	Irish Setter	cooperative	female	intact	5.5
18	Irish Setter	cooperative	female	spayed	5
19	Labrador Retriever	cooperative	female	intact	1.5
20	Labrador Retriever	cooperative	male	neutered	6
21	Lagotto Romagnolo	cooperative	male	neutered	7
22	Lagotto Romagnolo	cooperative	male	neutered	4
23	Hungarian Vizsla	cooperative	male	intact	1
24	Malinois	cooperative	female	intact	2
25	Puli	cooperative	female	spayed	9
26	Puli	cooperative	female	spayed	9
27	Pumi	cooperative	male	intact	4
28	Pumi	cooperative	female	spayed	1
29	Rottweiler	cooperative	female	spayed	5.5
30	Rottweiler	cooperative	male	intact	8
31	Smooth Collie	cooperative	male	intact	8
32	Smooth Collie	cooperative	male	intact	1.5
33	Smooth Collie	cooperative	female	intact	4
34	Shetland Sheepdog	cooperative	male	neutered	3
35	Rough Collie	cooperative	male	intact	5
36	Cardigan Welsh Corgi	cooperative	male	neutered	4
37	Airedale Terrier	independent	male	neutered	4
38	Airedale Terrier	independent	male	neutered	4
39	Akita Inu	independent	male	neutered	4
40	Basset Hound	independent	male	neutered	2
41	Basset Hound	independent	male	neutered	1
42	Basset hound	independent	female	spayed	2.5
43	Transylvanian Hound	independent	male	neutered	7
44	Transylvanian Hound	independent	male	neutered	4
45	Fox Terrier	independent	female	spayed	2.5
46	Hovawart	independent	male	neutered	3
47	Irish Terrier	independent	male	intact	4.5
48	Irish Terrier	independent	male	neutered	2
49	Irish Terrier	independent	female	spayed	5
50	Jack Russell Terrier	independent	male	intact	4.5
51	Jack Russell Terrier	independent	male	neutered	7
52	Jack Russell Terrier	independent	male	intact	3.5
53	Komondor	independent	female	intact	5
54	Borzoi	independent	male	intact	2
55	Borzoi	independent	male	intact	2
56	Borzoi	independent	female	intact	2
57	Parson Russell Terrier	independent	male	neutered	10
58	Rhodesian Ridgeback	independent	male	intact	4
59	Shiba Inu	independent	female	spayed	1
60	Shiba Inu	independent	male	neutered	1
61	Shiba Inu	independent	male	neutered	7
62	Shiba Inu	independent	male	neutered	3
63	Staffordshire Terrier	independent	male	neutered	8
64	Samoyed	independent	female	spayed	4
65	Dachshund	independent	male	neutered	1
66	Welsh Terrier	independent	male	neutered	11
67	Welsh Terrier	independent	male	neutered	4
68	Welsh Terrier	independent	female	intact	2.5
69	Whippet	independent	male	intact	2
70	Whippet	independent	male	intact	7
71	Whippet	independent	male	neutered	3

**Table 2 animals-14-02348-t002:** The list of behavioral variables we used during video coding. In the column of ‘behavioral variable’, we also indicated in which trial we coded the particular variable. Most of the listed variables did not occur (or occurred only very sporadically) in Trial 5 (i.e., in the familiarization phase, when the dogs could easily obtain the reward from the openable container).

Behavioral Variable	Unit	Description
First-touch latency (Trials 5 and 6)	(s)	The time elapsed between the moment of releasing the dog at the starting point and the moment when the dog touched the container (by paw, nose, or mouth) for the first time.
Reward latency (Trial 5 only)	(s)	The time elapsed between the moment of releasing the dog at the starting point and the when the dog took the reward from the container. In the case of an unsuccessful trial, 60 s was assigned.
Giving up latency (Trial 6 only)	(s)	The time elapsed between the moment of releasing the dog at the starting point and when the dog stopped going closer than 1 m to the container for 30 s. If the dog did not stop trying to get to the reward, the ‘giving up’ latency was 180 s.
Two-stage gaze alternation (Trial 6 only, frequency)	1/s	The dog either looked at the container first (by turning its head only or with full body orientation) and then immediately looked at the owner/experimenter, or vice versa. Number of gaze alternations divided by the giving up latency.
Interacting with the container with nose (Trials 5 and 6)	Relative duration	During attempts to get the reward from the container, the dog touched the container with its nose. Total time the dog spent with this behavior divided by the reward (Trial 5) or giving up latency (Trial 6).
Interacting with the container with paw (Trials 5 and 6)	Relative duration	During attempts to get the reward from the container, the dog touched the container with its paw. Total time the dog spent with this behavior divided by the reward (Trial 5) or giving up latency (Trial 6).
Looking at the container (Trials 5 and 6)	Relative duration	The total time the dog spent looking at the container (by turning its head only or with full body orientation) divided by the reward (Trial 5) or giving up latency (Trial 6).
Task focus (Trial 6 only)	Relative duration	Task focus describes the dogs’ tendency to leave the close vicinity of the container. The dog stepped over the boundary line drawn 1 m from the container while moving back toward the owner. The value was calculated from the total time the dog spent away from the container, divided by the giving up latency.
Looking back (Trial 6 only, frequency)	1/s	The dog turned toward the owner/experimenter (by turning its head only or with full body orientation) and looked at them during its attempts to get the reward from the container. Number of looking back events divided by the giving up latency.
Encouragement by the owner (Trial 6 only, frequency)	1/s	The number of distinct verbal utterances (at least 1 s between two) given by the owner during the dog’s attempts to get the reward from the container, divided by the giving up latency

**Table 3 animals-14-02348-t003:** Results of Generalized Linear Model analyses in the case of the 5th trial of the familiarization phase for looking at the container and interacting with their nose. The table shows the Benjamini–Hochberg adjusted *p*-values.

Dependent Variable	Factor	Chi-Square	Df	*p*
Looking at the container	Breed group	1.422	1	0.291
	Training	1.437	1	0.291
	Keeping	4.571	1	0.161
	Sex	0.440	1	0.507
	Reproductive status	3.331	1	0.170
Interacting with the container with nose	Breed group	0.924	1	0.562
	Training	0.245	1	0.621
	Keeping	2.548	1	0.530
	Sex	0.434	1	0.621
	Reproductive status	1.555	1	0.530

**Table 4 animals-14-02348-t004:** Results of Generalized Linear Model analyses in the case of the test phase for frequency of LBR. The table shows the Benjamini–Hochberg adjusted *p*-values. Significant associations are highlighted with bold letters.

Dependent Variable	Factor	Chi-Square	Df	*p*
Frequency of LBR	**Breed group**	**9.310**	**1**	**0.012**
	Training	0.076	1	0.783
	Keeping	0.502	1	0.719
	Sex	0.164	1	0.783
	Reproductive status	0.621	1	0.719

**Table 5 animals-14-02348-t005:** Results of Generalized Linear Model analyses in the case of the test phase for the frequency of two-stage gaze alternation (TSGA). The table shows the Benjamini–Hochberg adjusted *p*-values. Significant associations are highlighted with bold letters.

Dependent Variable	Factor	Chi-Square	Df	*p*
Frequency of TSGA	**Breed group**	**10.604**	**1**	**0.005**
	Training	3.814	1	0.128
	Keeping	0.270	1	0.603
	Sex	1.422	1	0.388
	Reproductive status	0.561	1	0.568

**Table 6 animals-14-02348-t006:** Results of Cox regression analysis in the case of the test phase for first-touch latency. Significant associations are highlighted with bold letters.

Dependent Variable	Factor	Chi-Square	Df	*p*
First-touch latency	Group	2.480	1	0.115
	**Keeping**	**7.136**	**1**	**0.008**
	Training level	8.204	4	0.081

**Table 7 animals-14-02348-t007:** Results of Generalized Linear Model analyses in the case of the test phase for the following motivation-related behaviors: looking at the container, task focus duration, interactions with the container, and owner encouragement. The table shows the Benjamini–Hochberg adjusted *p*-values. Significant associations are highlighted with bold letters.

Dependent Variable	Factor	Chi-Square	Df	*p*
Looking at the container	Breed group	0.071	1	0.908
	Training	1.544	1	0.535
	Keeping	2.509	1	0.535
	Sex	0.013	1	0.908
	Reproductive status	0.460	1	0.828
Task focus duration	Breed group	0.090	1	0.955
	Training	1.321	1	0.793
	Keeping	0.769	1	0.793
	Sex	0.000	1	0.987
	Reproductive status	0.509	1	0.793
Owner encouragement	Breed group	0.965	1	0.817
	Training	0.000	1	0.989
	Keeping	0.476	1	0.817
	Sex	0.798	1	0.817
	Reproductive status	0.092	1	0.953
Interacting with the container with nose	Breed group	0.127	1	0.721
	Training	0.188	1	0.721
	Keeping	0.196	1	0.721
	Sex	2.431	1	0.595
	Reproductive status	0.601	1	0.721
Interacting with the container with paw	**Breed group**	**5.882**	**1**	**0.045**
	Training	0.487	1	0.582
	Keeping	4.598	1	0.064
	**Keeping–breed group interaction**	**6.151**	**1**	**0.045**
	Sex	1.101	1	0.441
	Reproductive status	0.001	1	0.976

## Data Availability

The dataset used for the analyses is provided as Appendix A.

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
