# Peer review of "Cooperative but Dependent–Functional Breed Selection in Dogs Influences Human-Directed Gazing in a Difficult Object-Manipulation Task"

_animals, 2024, doi:10.3390/ani14162348_

Round 1

Reviewer 1 Report

Comments and Suggestions for Authors

The aim of the study was to test the hypothesis that ‘cooperative’ dog breeds would also show higher levels of dependency towards humans in a difficult problem-solving task, than dogs that were selected for being independent problem solvers. For this purposes researchers examined such beahaviour as ‘looking back at the human’, which is a typical reaction of socialized dogs to an ‘unsolvable task’. It was tested dogs breeds from the independent (N=35) and cooperative working dog groups(N=36) and compared their results using different bahavioral variables.

Manuscript sent to me for review is clear and relevant for the field. The structure of the article is correct, the methods and results described in a logical and clear manner. Experimental design is appropriate to test the hypothesis. The statistical analysis is presented in detail. All figures and tables properly show the data, and they are easy to interpretation. Manuscript is well written, introduction is very interesting and contains rich background information. Discussion is clear and logically constructed. The conclusions are consistent with the results of experiment.

The change in dog behavior during the domestication process and the subsequent impact of artificial selection by humans on the development of characteristics specific to a given breed/breed group of dogs is still a very important topic of research. There are still many questions related to this issue. Therefore, any research that increases knowledge about the impact of artificial selection on behavior and confirms or denies actual differences between breeds is valuable and worth publishing.

The only weak point of this manuscript is the lack of an adequate interpretation of the results related to the variable "keeping condition". The interpretation of the influence of variable - indoor only or outdoor access - on the dog's behavior during the performance of the unsolvable task is superficial and unconvincing. It is worth trying to develop this thread in the discussion.

Specific comments:

Line 213 – “only purebred” - What canine organization was taken into account? FCI or also other organizations?

Table 1 - Is such a detailed table necessary in the text of the publication? Can it be published in a shortened form and in its entirety only as supplementary material? This is only my suggestion and I leave the final decision to the Editor.

Table 3, 4, 5, 6, 7 – bold letters are invisible

Figure 3 - The manuscript contains enough graphs and tables that Figure 3 seems to me unnecessary  but and I leave the final decision to the Editor.

Author Response

REVIEWER#1

The aim of the study was to test the hypothesis that ‘cooperative’ dog breeds would also show higher levels of dependency towards humans in a difficult problem-solving task, than dogs that were selected for being independent problem solvers. For this purposes researchers examined such beahaviour as ‘looking back at the human’, which is a typical reaction of socialized dogs to an ‘unsolvable task’. It was tested dogs breeds from the independent (N=35) and cooperative working dog groups(N=36) and compared their results using different bahavioral variables.

Manuscript sent to me for review is clear and relevant for the field. The structure of the article is correct, the methods and results described in a logical and clear manner. Experimental design is appropriate to test the hypothesis. The statistical analysis is presented in detail. All figures and tables properly show the data, and they are easy to interpretation. Manuscript is well written, introduction is very interesting and contains rich background information. Discussion is clear and logically constructed. The conclusions are consistent with the results of experiment.

RESPONSE: We are thankful for the supportive comments and appreciate the thoughtful questions/criticism of the Reviewer. We adjusted the manuscript accordingly to your suggestions.

The change in dog behavior during the domestication process and the subsequent impact of artificial selection by humans on the development of characteristics specific to a given breed/breed group of dogs is still a very important topic of research. There are still many questions related to this issue. Therefore, any research that increases knowledge about the impact of artificial selection on behavior and confirms or denies actual differences between breeds is valuable and worth publishing.

RESPONSE: Thank you for the commending words. We feel that exploring the potential effects of functional breed selection on dogs’ behavior has a lot of new discoveries still to come.

The only weak point of this manuscript is the lack of an adequate interpretation of the results related to the variable "keeping condition". The interpretation of the influence of variable - indoor only or outdoor access - on the dog's behavior during the performance of the unsolvable task is superficial and unconvincing. It is worth trying to develop this thread in the discussion.

RESPONSE: Thank you for highlighting the need for more elaboration here. We added a longer section to the Discussion (lines 461-477), where we examined two possible explanations:

This last result was independent of the breed group-designation of the subjects. The original task of many breeds included the need for touching/grabbing of live or dead animals, and these breeds were represented in both breed groups (e.g., retrievers, pointers and utility breeds in the ‘cooperative’ group; and terriers, sighthounds in the ‘independent’ group). We could therefore speculate that the difference between the first touch latencies of subjects with regard to their keeping conditions could be connected to the various levels of novelty they normally experience at home. As dogs with free access to the outdoors probably encounter more novel stimuli on a regular basis, they may react with less enthusiasm to the objects provided during a test. Indoor-only dogs on the other hand could take the test with the container and reward as a case of environmental enrichment, and reacted with higher motivation. Another explanation could be that the owners of indoor-only dogs may provide their dogs with puzzle-games more often than the indoor-outdoor dogs’ owners do. Therefore, indoor-only dogs could be more familiar with these types of problem-solving tasks and approached the container faster in our test, too.

Specific comments:

Line 213 – “only purebred” - What canine organization was taken into account? FCI or also other organizations?

RESPONSE: We considered any dog breed that is recognized by one or the other major kennel clubs (FCI, AKC, UKC, KC). We added this information to the text (lines 215-216).

Table 1 - Is such a detailed table necessary in the text of the publication? Can it be published in a shortened form and in its entirety only as supplementary material? This is only my suggestion and I leave the final decision to the Editor.

RESPONSE: We think the basic details of the participating dogs are important and can be interesting for the readers. If the Editor agrees, we would keep this table in the text, but of course, we are not against moving it to the Supplementary section either.

Table 3, 4, 5, 6, 7 – bold letters are invisible

RESPONSE: Thank you for the comment, we corrected the tables.

Figure 3 - The manuscript contains enough graphs and tables that Figure 3 seems to me unnecessary  but and I leave the final decision to the Editor.

RESPONSE: We agree with the Reviewer that these Figures were not very informative. We deleted them.

Reviewer 2 Report

Comments and Suggestions for Authors

Seeking for advice from handler by looking  back at him/her in case of unsolvable or difficult tasks is an interesting  phenomenon in domestic dogs not only theoretically, from the point of view of domestication and selective breeding history, but also from practical point of view e.g. in scent dogs.

The authors compared 18 „cooperative” and 18 „independent” dog breeds for their level of interactivity with 14 their handlers during solvable and unsolvable task. The authors assumed that  functional breed selection in post-domestication period of dog breeds resulted in behavioral phenotypes that differentially relate  to dog-human interactions in this case in form of „looking back” at a human when the task for a dog is unsolvable. The distinction between cooperative and independent breeds adopted by the authors seems to be based on common opinions and in case of some breeds could be disputable. It is known that within some breeds so called working  lines that are more or less systematically subjected to various performance tests. On the other hand within „cooperative” breeds so called  show lines that are selected only on appearance, exists. In some breeds either working or show lines prevail. This should be taken into consideration. In many cases the dog breeds’ original function and currently characteristic  diverged due to genetic relatedness, typical purpose and keeping conditions. This may result in some breeds being hard-to-classify as to their „cooperativeness” and/or „independency”.  The authors are correct when  suggesting that  „..for investigating the effect of functional working breed selection on the dependency of dogs on human assistance, a much clearer and more representative choice of breeds would be needed…”. It is worth mentioning that the LBR is only one of specific measures for cooperativeness and in my opinion does not exhaust the notion of „cooperativeness”.  Interestingly in some types of tasks for scent dogs belonging to „cooperative” breeds, a moderate LBR when searching in field setting may be  advatageous, because the handler could better encourage the dog for a more systematic search (provided the handler is blind to the location of the target scent). On the contrary, in  scent dogs working in a scent lineup, a tendency for the LBR is disadvatageous since it may result in „clever Hans effect” if a rigorous double blind protocol is not applied, which, unfortunately,  practically is sometimes a case.

The authors informed that they were „paying careful attention to the balanced distribution of sex, age, keeping condition and training level”, however, I guess that the distribution of 71 dogs for 2 behavioural categories, 2 sexes, 2 reproductive statuses, 3 keeping conditions and 6 training leves, might result in a very small number of dogs in each of the category or no dogs in some categories. This migt be a cause for no significant association between the frequency of LBR and potential confounders. This question should be addressed more explicitly.

Author Response

REVIEWER#2

Comments and Suggestions for Authors

Seeking for advice from handler by looking back at him/her in case of unsolvable or difficult tasks is an interesting phenomenon in domestic dogs not only theoretically, from the point of view of domestication and selective breeding history, but also from practical point of view e.g. in scent dogs.

RESPONSE: Thank you for the nice recapping of the main topic of the paper. We are in general thankful for the supportive comments and interesting suggestions from the Reviewer. We did our best to incorporate these into the manuscript.

The authors compared 18 „cooperative” and 18 „independent” dog breeds for their level of interactivity with their handlers during solvable and unsolvable task. The authors assumed that functional breed selection in post-domestication period of dog breeds resulted in behavioral phenotypes that differentially relate to dog-human interactions in this case in form of „looking back” at a human when the task for a dog is unsolvable.

RESPONSE: Thank you for the summary of the research goal of the paper.

 The distinction between cooperative and independent breeds adopted by the authors seems to be based on common opinions and in case of some breeds could be disputable. It is known that within some breeds so called working lines that are more or less systematically subjected to various performance tests. On the other hand within „cooperative” breeds so called show lines that are selected only on appearance, exists. In some breeds either working or show lines prevail. This should be taken into consideration. In many cases the dog breeds’ original function and currently characteristic diverged due to genetic relatedness, typical purpose and keeping conditions. This may result in some breeds being hard-to-classify as to their „cooperativeness” and/or „independency”.

RESPONSE: We decided upon the type (cooperative or independent) of the breeds based on the original working function taken from the official breed descriptions. This detail we have specified in the text (lines 211-223). The Reviewer is right in the mentioned issues, and we agree with the Reviewer that purebred dogs that are commonly kept as companions nowadays are almost surely diverged from the landraces/ intentional breed-crosses they are originating from. We also agree that the division of breeds to ‘working’ and ‘show’ lines resulted in more and more differently behaving versions of the same breed (e.g., Sundman et al., 2016). The dogs in our sample were generally from the ‘pet-population’ as we haven’t been specifically recruiting participants neither from social media communities of dog shows or working-dog enthusiasts. Now we included this notion to the Limitations of the study (lines 600-609), and it reads like this:

Another limitation is that we sorted the various working dog breeds to the ‘independent’ or ‘cooperative’ group based on their original task taken from the official breed standards. However, recent preferences in dog breeding for either more family-friendly, or work-oriented behavior traits resulted in the development of genetically and behaviorally different (Fadel et al., 2016; Sundman et al., 2016) ‘lines’ within some breeds (e.g., Retrievers, Border Collies, Australian Kelpies etc.). As we used only a few specimens from each breed to our study, we could not balance between the potentially different within-breed variants – although based on our recruiting efforts, we can assume that our subjects were family companion dogs.

Another way we were trying to avoid serious biases caused by the unwanted participation of large numbers of line-selected subjects was the inclusion of as many breeds/breed type as possible with only a few specimens per breed. Thus, even if we would have some line-selected dogs in our sample, their influence could be minimal. We intentionally avoided the inclusion of those breeds where the original work function has been long ago taken out from the goals of breeding (such as the English Bulldog), or the breed had clearly a ‘toy/lapdog’ role. We added this detail to the Methods (lines 219-223), where it reads like this:

We intentionally avoided the inclusion of those breeds where, the original work function has been long ago taken out from the goals of breeding (such as the English Bulldog), or the breed had clearly a ‘toy/companion’ designation (such as the Pekingese or the Bolognese).

Finally, our main results showed the expected difference between the independent and cooperative dogs according to our hypothesis. This provides support to the validity of our method of group-formation (see also the similarly assembled test groups at Gácsi et al., 2009; Pongrácz et al., 2021; Dobos and Pongrácz, 2023, 2024; Lugosi et al., 2024).

 The authors are correct when suggesting that „..for investigating the effect of functional working breed selection on the dependency of dogs on human assistance, a much clearer and more representative choice of breeds would be needed…”. It is worth mentioning that the LBR is only one of specific measures for cooperativeness and in my opinion does not exhaust the notion of „cooperativeness”. Interestingly in some types of tasks for scent dogs belonging to „cooperative” breeds, a moderate LBR when searching in field setting may be advatageous, because the handler could better encourage the dog for a more systematic search (provided the handler is blind to the location of the target scent). On the contrary, in scent dogs working in a scent lineup, a tendency for the LBR is disadvatageous since it may result in „clever Hans effect” if a rigorous double blind protocol is not applied, which, unfortunately, practically is sometimes a case.

RESPONSE: Thank you for this thoughtful comment. The difference between the advantageousness of LBR depending on the nature of scent-task is definitely interesting and new to us. Definitely, the unsolvable task/LBR is just one of those human interaction-based scenarios where the independent and cooperative breeds can be expected behave differently (e.g., following visual pointing Gácsi et al., 2009, social learning from humans (Dobos and Pongrácz, 2023) and dogs (Lugosi and Pongrácz, 2024), relying on human verbal communication etc.). In our opinion, differences in LBR between the cooperative and independent breeds taps more in the different levels of human-dependency of these breeds than their actual cooperativity. We incorporated the Reviewer’s idea to the closing section of the Conclusions, (lines 635-639), where it reads like this:

“In the future, it would be interesting to investigate whether the stronger dependency of cooperative dog breeds on humans actually would enhance or lessen their work efficiency in realistic circumstances, such as scent detection work. Stronger dependency could mean higher sensitivity to human-given cues, which can theoretically lead unwanted bias in scent detecting scenarios [Lazarowski et al., 2019].”

The authors informed that they were „paying careful attention to the balanced distribution of sex, age, keeping condition and training level”, however, I guess that the distribution of 71 dogs for 2 behavioural categories, 2 sexes, 2 reproductive statuses, 3 keeping conditions and 6 training leves, might result in a very small number of dogs in each of the category or no dogs in some categories. This migt be a cause for no significant association between the frequency of LBR and potential confounders. This question should be addressed more explicitly.

RESPONSE: The Reviewer is right: with the available number of subjects, we could not have a representative size subsample from each combination of variables. Our goal was to create a representative enough sample for the two breed groups, and additionally, keeping the potential confounders equally distributed between the two breed groups (i.e., avoiding that independent and cooperative dogs would come in markedly different ratio of sex, keeping conditions and training levels). This is a limitation of the study and now we mention it in the manuscript (lines 610-616).

Finally, while our sample was representative and balanced from the aspect of cooperative and independent working dog breeds, with the limited number of subjects it was not possible to provide a large enough subsample of each combination of confounders (sex, reproductive status, keeping and training conditions). As our main goal was to test the potential association between functional breed selection and looking-back response of dogs in a difficult task, we aimed at having similar distribution of confounders across the two breed groups.

Reviewer 3 Report

Comments and Suggestions for Authors

Review of Pongrácz and Lugosi’s Cooperative but dependent – functional breed selection in dogs influences human-directed gazing in an unsolvable object-manipulation task, manuscript id animals-3140581

Pongrácz and Lugosi’ address an interesting question – do “cooperative” dog breeds and “independent” dog breeds respond differently when given a difficult task.  The introduction nicely develops their reasoning for why dogs in the two groups might behave differently.  The method leaves out important details such as how breeds were classified as cooperative vs independent.  The results section presents the inferential statistics from many tests but does not address how Type I errors are controlled in the family of tests.  In the discussion the authors point out that the results cannot be explained by several possible confounding variables.  Given the quasi-experimental nature of their design, other potential confounds cannot be ruled out.

Issues:

Paragraph starting at line 212:  “Cooperative” and “independent” breeds are fundamental to your study.  It would be helpful if you explained how you classified particular breeds as cooperative or as independent.  Is there a source that you can cite for your classification scheme?

Paragraph starting at line 212:  You treat “cooperative” and “independent” breeds as a dichotomy – two groups with nothing in-between.  Is that the case?  Or, are cooperative and independent two end-points of a continuous variable?  If some breeds in each group are close to the mid-point of the continuum, this might partially explain why you sometimes do not find statistically significant results.

Table 2:   You have 10 dependent variables (some of which were not analyzed) and performed many inferential statistical tests.  While your α level (which is not specified in the manuscript) helps protect each individual test from Type I errors, the more tests that you perform, the more likely that at least one of them will result in a Type I error.  What did you do to protect yourself from making Type I errors across the family of inferential statistical tests?

Lines 245, 462, and 612:  This study was not an experiment in the sense that nothing was manipulated and participants were not randomly assigned to conditions.  Rather, dogs were assigned to conditions based on attributes of the dogs.  “Quasi-experiment” would be a more appropriate description of your design.  As such, you can never be certain that the factor (breed group) is the cause of the change in the dependent variable.  There is always a possibility that something that co-varies with the factor is causing the change in the dependent variables.  While you rule out several important confounds, it is impossible to rule all such potential confounds.  How does this affect your conclusions?

Minor Issues:

Line 25:  “for the behavioral differences in dogs.”  Which behavioral differences are you talking about?  All of them?  Or, is it just the “looking back at the human” behavior as suggested by the next sentence?  Perhaps, “was responsible for differences in whether dogs look back at a human when presented with an unsolvable task” would be better with a subsequent modification in the next sentence to remove the redundancy.

Line 29: “from18-18 cooperative or independent breeds” is awkward.  Do you mean “from 18 cooperative and 18 independent breeds”?

Line 31: Was the task really impossible?  No dog, no matter how much time they were given, could ever unlock the container or bite through it?  You use the term “difficult” to describe the task on line 15 and that may be a more appropriate term to use throughout the manuscript, including the title.

Line 35:  As you are undoubtedly aware, inferential statistics do not show a lack of association.  Rather, when you fail to reject the null, they fail to show an association which may be due to a lack of association, a Type II error, low statistical power, violations of assumptions, etc.

Line 41:  As you are undoubtedly aware, inferential statistics are not proof of anything.  They make a probabilistic statement that the observed difference or association is not due to chance.  In statistics, the conclusion can be wrong – a Type I or Type II error might occur.  In a proof, the conclusion must be correct as long as the assumptions are correct.

Line 63:  Change “Additionally, (compared to wolves) dogs…” to “Additionally, compared to wolves, dogs…”

Line 74:  Include a period at the end of the sentence.

Line 80:  Insert a space between the end of the sentence and the start of the next sentence.

Line 83:  The closing parenthesis is missing.

Line 220: Similar to the comment at line 29.

Line 272:  Since the reward could be food or a toy (line 240), “food” should be replaced with “reward”.

Paragraph starting at line 274: “E” and “O” are awkward – replace with experimenter and owner.

Line 277: Would “explore” be a better word than “discover”?

Tables 3 through 7:  The table captions state that “significant effects are highlighted with bold letters” but the body of the tables do not appear to have any bold letters other than the column headings.

Tables 3 through 7:  These are not independent variables in the sense that you did not systematically manipulate the variables.  Rather, dogs were assigned to the levels of the factor based on attributes of the dogs.  Replace “independent variable” with either “factor” or “quasi-independent variable”.

Figures 1 and 2: What do the “whiskers” in the box plot represent?  95% confidence interval, standard deviation, standard error, range, interquartile range?

Figures:  The figures would look more professional if “breedgroup” was replaced with “Breed group” and the legend had word labels (“Trained at home only”) in place of number labels (“2”).  “Latency_firsttouch” should be “Latency to first touch”.

Line 597:  Should “Funding:” be in a bold font with a blank line between lines 596 and 597?

Line 669:  The reference is incomplete and “SCIentIfIC REPOrTS” is strange.

Comments on the Quality of English Language

In general, the quality of English is fine.  There are several minor corrections to make.

Author Response

REVIEWER #3

Comments and Suggestions for Authors

Review of Pongrácz and Lugosi’s Cooperative but dependent – functional breed selection in dogs influences human-directed gazing in an unsolvable object-manipulation task, manuscript id animals-3140581

Pongrácz and Lugosi’ address an interesting question – do “cooperative” dog breeds and “independent” dog breeds respond differently when given a difficult task. The introduction nicely develops their reasoning for why dogs in the two groups might behave differently. The method leaves out important details such as how breeds were classified as cooperative vs independent. The results section presents the inferential statistics from many tests but does not address how Type I errors are controlled in the family of tests. In the discussion the authors point out that the results cannot be explained by several possible confounding variables. Given the quasi-experimental nature of their design, other potential confounds cannot be ruled out.

RESPONSE: Thank you for the supportive opinion and for the helpful criticism. Here we summarize the main changes, below we respond in more detailed manner.

We added further details to the text about how the breeds were selected, which breeds have been excluded. We also expanded the Limitations of the study, based on your comments about the potential problems with breeds that recently started to diverge towards distinct ‘lines’ due to different goals of selection.

Following the Reviewer’s advice, now we applied the Benjamini-Hochberg correction to the p-values in most models and we reported the new values in the revised manuscript.

Issues:

Paragraph starting at line 212: “Cooperative” and “independent” breeds are fundamental to your study. It would be helpful if you explained how you classified particular breeds as cooperative or as independent. Is there a source that you can cite for your classification scheme?

RESPONSE: Thank you for the question – this is truly an important detail. When we classified the dog breeds into the ‘independent’ or ‘cooperative’ group, we relied on the official descriptions from the breed standards. We avoided using such dog breeds that were not working breeds but they were used for toy/companion purposes (‘lapdogs’). The same method of functional breed categorization can be found in a number of earlier publications (Gácsi et al., 2009; Bognár et al., 2022; Dobos and Pongrácz, 2023). The text section about these details was rewritten and enhanced in the Materials and Methods chapter (lines 211-223), now it reads like this:

To categorize the dog breeds according to their original work functions, we followed the method that was used in a number of earlier publications (e.g., [6,9,12]). Only purebred dogs were recruited that belonged to either the cooperative working breeds, or to the independent working breeds. We considered any dog breed that is recognized by one or the other major kennel clubs (FCI, AKC, KC). Only those breeds were used where their function type (independent or cooperative) was possible to be unambiguously determined according to the historical breed description, accessible from the official breed standards. We intentionally avoided the inclusion of those breeds where, the original work function has been long ago taken out from the goals of breeding (such as the English Bulldog), or the breed had clearly a ‘toy/companion’ designation (such as the Pekingese or the Bolognese).

Paragraph starting at line 212: You treat “cooperative” and “independent” breeds as a dichotomy – two groups with nothing in-between. Is that the case? Or, are cooperative and independent two end-points of a continuous variable? If some breeds in each group are close to the mid-point of the continuum, this might partially explain why you sometimes do not find statistically significant results.

RESPONSE: Thank you for pointing out this important limitation of using of the classic functional descriptions of dog breeds for categorizing them. Our method (just like in the other, earlier papers from the similar genre) was a quasi-binary one: working dog breeds were either classified as ‘independent’ or ‘cooperative’. However, to lessen the ambiguity, we excluded such breeds that had no clear task according to their descriptions apart from being ‘companions’ (the so-called ‘toy category’). We agree with Reviewer that particular dog breeds probably still had not their designated group, we tried to minimize the resulting bias with using as many breeds/group as possible and use only a few dogs per breed. Ideally, dog breeds should be placed to an independence-cooperativity scale – an intriguing opportunity for the near future follow-up research. We considerably enhanced the Limitations section at the end of the Discussion, where now we mention the problems with breed classifications, for example because of within-breed differentiation between working and ‘show’ lines with sometimes markedly different behavioral traits (lines 591-616).

Table 2: You have 10 dependent variables (some of which were not analyzed) and performed many inferential statistical tests. While your α level (which is not specified in the manuscript) helps protect each individual test from Type I errors, the more tests that you perform, the more likely that at least one of them will result in a Type I error. What did you do to protect yourself from making Type I errors across the family of inferential statistical tests?

RESPONSE: Thank you for this comment. Now we added to the statistical analysis chapter that the α level was 0.05. There are two variables in Table 2 (latency of obtaining the reward in Trial 5, and latency of giving up in Trial 6), which were indeed not analyzed on their own, however, we used these for calculating relative frequencies and durations. This is why we thought that their description should also be in the table.

Following the Reviewer’s advice, we applied the Benjamini-Hochberg adjustment to the p-values in the case of most analyses, where we had 5-6 factors per model (lines 330-332). In the Tables we report the adjusted p-values now. With this procedure, we still have the significant effects of the breed groups, but some of the significant interactions have disappeared – the Discussion and Abstract were also amended accordingly.

Lines 245, 462, and 612: This study was not an experiment in the sense that nothing was manipulated and participants were not randomly assigned to conditions. Rather, dogs were assigned to conditions based on attributes of the dogs. “Quasi-experiment” would be a more appropriate description of your design. As such, you can never be certain that the factor (breed group) is the cause of the change in the dependent variable. There is always a possibility that something that co-varies with the factor is causing the change in the dependent variables. While you rule out several important confounds, it is impossible to rule all such potential confounds. How does this affect your conclusions?

RESPONSE: Thank you for this interesting comment, we definitely did not think that our study would be a quasi-experiment (indeed, this was the first time we read about this term). We would argue that changing the container from openable to non-openable between Trials 5 and 6 could be considered as “experimental manipulation”. We agree that there can be additional confounders. In our opinion the most relevant confounding behavioral traits, which we definitely could not rule out were the more specific work-related behavioral characteristics in particular breeds. We suspect that for example, the inclination for digging in terriers and Dachshunds (dog breeds that were bred for underground hunting), could be the reason that we found more frequent ‘pawing’ in the indoor-only independent dogs. We mention this in the Limitations section (lines 593-600). As for the terminology, we followed the advice of Reviewer, and changed “experiment” to “investigation” and “experimental groups” to “test groups” in our manuscript.

Minor Issues:

Line 25: “for the behavioral differences in dogs.” Which behavioral differences are you talking about? All of them? Or, is it just the “looking back at the human” behavior as suggested by the next sentence? Perhaps, “was responsible for differences in whether dogs look back at a human when presented with an unsolvable task” would be better with a subsequent modification in the next sentence to remove the redundancy.

RESPONSE: Thank you for calling our attention to this ambiguously worded sentence. We rewrote it (lines 24-26), now it reads:

It is still largely unknown, to which extent domestication, ancestry or recent functional selection was responsible for the behavioral differences in whether dogs look back to a human when presented with a difficult task. Here we tested whether this ubiquitous human-related response of companion dogs would appear differently in subjects that were selected for either cooperative or independent work tasks.

Line 29: “from18-18 cooperative or independent breeds” is awkward. Do you mean “from 18 cooperative and 18 independent breeds”?

RESPONSE: Thank you for the suggestion, we changed the text accordingly.

Line 31: Was the task really impossible? No dog, no matter how much time they were given, could ever unlock the container or bite through it? You use the term “difficult” to describe the task on line 15 and that may be a more appropriate term to use throughout the manuscript, including the title.

RESPONSE: The Reviewer is right. Although it happened only twice, exceptionally motivated and destructive dogs managed to break the plastic container (although they eventually did not obtain the reward). Still, we agree that we should rather call the task as “difficult” – so we have changed the title of the article accordingly, and also swapped “unsolvable” to “difficult” at most places in the text. At a few places, we left “unsolvable phase” in the text, because this is how traditionally researchers distinguish between the ‘solvable’ and ‘difficult’ phases in these test scenarios.

Line 35: As you are undoubtedly aware, inferential statistics do not show a lack of association. Rather, when you fail to reject the null, they fail to show an association which may be due to a lack of association, a Type II error, low statistical power, violations of assumptions, etc.

RESPONSE: We use a different wording now, hopefully overcoming the problem what the Reviewer highlighted here:

Task focus, looking at the container, touching the container for the first time, or interacting with the container with a paw or nose, did not differ between the breed groups…

Line 41: As you are undoubtedly aware, inferential statistics are not proof of anything. They make a probabilistic statement that the observed difference or association is not due to chance. In statistics, the conclusion can be wrong – a Type I or Type II error might occur. In a proof, the conclusion must be correct as long as the assumptions are correct.

RESPONSE: Thank you for the comment, we agree. We replaced the word “prove” with “suggest”:

These are the first empirical results that suggest in a balanced, representative sample of breeds, that the selection for different levels of cooperativity in working dogs could also affect their human-dependent behavior in a generic problem-solving situation.

Line 63: Change “Additionally, (compared to wolves) dogs…” to “Additionally, compared to wolves, dogs…”

RESPONSE: Done.

Line 74: Include a period at the end of the sentence.

Line 80: Insert a space between the end of the sentence and the start of the next sentence.

Line 83: The closing parenthesis is missing.

Line 220: Similar to the comment at line 29.

Line 272: Since the reward could be food or a toy (line 240), “food” should be replaced with “reward”.

Paragraph starting at line 274: “E” and “O” are awkward – replace with experimenter and owner.

Line 277: Would “explore” be a better word than “discover”?

RESPONSE: All the above suggested edits were performed according to the recommendation of the Reviewer.

Tables 3 through 7: The table captions state that “significant effects are highlighted with bold letters” but the body of the tables do not appear to have any bold letters other than the column headings.

REPLY: Thank you for catching this mistake – now we truly highlighted the significant associations in the tables.

Tables 3 through 7: These are not independent variables in the sense that you did not systematically manipulate the variables. Rather, dogs were assigned to the levels of the factor based on attributes of the dogs. Replace “independent variable” with either “factor” or “quasi-independent variable”.

RESPONSE: Instead “independent variable”, we put “factor” to the table headings.

Figures 1 and 2: What do the “whiskers” in the box plot represent? 95% confidence interval, standard deviation, standard error, range, interquartile range?

RESPONSE: The missing explanations were added to the figure legends:

“Whiskers: minimum, maximum; box: Q1-Q3 interquartile range; horizontal line: median”

Figures: The figures would look more professional if “breedgroup” was replaced with “Breed group” and the legend had word labels (“Trained at home only”) in place of number labels (“2”). “Latency_firsttouch” should be “Latency to first touch”.

RESPONSE: We have deleted the less informative Figure 3, and the other Figures were replaced with better-drawn versions. We put emphasis on the more professional look at this time.

Line 597: Should “Funding:” be in a bold font with a blank line between lines 596 and 597?

RESPONSE: Corrected, thank you.

Line 669: The reference is incomplete and “SCIentIfIC REPOrTS” is strange.

RESPONSE: Corrected, thank you

Round 2

Reviewer 3 Report

Comments and Suggestions for Authors

The authors have addressed my concerns.  Thank you.

References 47 and 48 are formatted differently than the other references -- the second lines are not indented.